# SparsePro: An efficient fine-mapping method integrating summary statistics and functional annotations

Wenmin Zhang[1]*, Hamed Najafabadi[1,2,3], Yue Li[1,4]*

**1** Quantitative Life Sciences, McGill University, Montreal, Quebec, Canada, **2** Department of Human Genetics, McGill University, Montreal, Quebec, Canada, **3** Dahdaleh Institute of Genomic Medicine, Montreal, Quebec, Canada, **4** School of Computer Science, McGill University, Montreal, Quebec, Canada

* wenmin.zhang@mail.mcgill.ca (WZ); yueli@cs.mcgill.ca (YL)

**Data Availability Statement:** SparsePro is an open-access software publicly available at \url {https://github.com/zhwm/SparsePro}. The simulation scripts are deposited at \url

## Abstract

Identifying causal variants from genome-wide association studies (GWAS) is challenging due to widespread linkage disequilibrium (LD) and the possible existence of multiple causal variants in the same genomic locus. Functional annotations of the genome may help to prioritize variants that are biologically relevant and thus improve fine-mapping of GWAS results. Classical fine-mapping methods conducting an exhaustive search of variant-level causal configurations have a high computational cost, especially when the underlying genetic architecture and LD patterns are complex. SuSiE provided an iterative Bayesian stepwise selection algorithm for efficient fine-mapping. In this work, we build connections between SuSiE and a paired mean field variational inference algorithm through the implementation of a sparse projection, and propose effective strategies for estimating hyperparameters and summarizing posterior probabilities. Moreover, we incorporate functional annotations into fine-mapping by jointly estimating enrichment weights to derive functionally-informed priors. We evaluate the performance of SparsePro through extensive simulations using resources from the UK Biobank. Compared to state-of-the-art methods, SparsePro achieved improved power for fine-mapping with reduced computation time. We demonstrate the utility of SparsePro through fine-mapping of five functional biomarkers of clinically relevant phenotypes. In summary, we have developed an efficient fine-mapping method for integrating summary statistics and functional annotations. Our method can have wide utility in understanding the genetics of complex traits and increasing the yield of functional follow-up studies of GWAS. SparsePro software is available on GitHub at https://github.com/zhwm/SparsePro.

## Author summary

Accurately identifying causal variants from genome-wide association studies summary statistics is important for understanding genetic architecture of complex traits and identifying therapeutic targets. Functional annotations are commonly used as additional evidence for prioritizing causal variants. In this study, we present SparsePro to integrate

{https://github.com/zhwm/SparsePro_analysis}. Individual-level phenotype and genotype data from the UK Biobank are available upon successful application at \url{https://www.ukbiobank.ac.uk}. GCTA was downloaded from \url{https://cnsgenomics.com/software/gcta/bin/gcta_1.93.2beta.zip}. FINEAMP was downloaded from \url{http://www.christianbenner.com/finemap_v1.4_x86_64.tgz}. SuSiE (version 0.12.16) was installed from CRAN. PolyFun was installed from \url{https://github.com/omerwe/polyfun}. UK Biobank LD information was downloaded from \url{https://alkesgroup.broadinstitute.org/UKBB_LD/}. Tissue-specific annotation was downloaded from \url{https://alkesgroup.broadinstitute.org/LDSCORE/}.

**Funding:** W.Z. has been supported by a doctoral training fellowship from the FRQNT (319188) and the Healthy Brains, Healthy Lives Program, funded by the Canada First Research Excellence Fund (CFREF), Quebec's Ministère de l'Économie et de l'Innovation (MEI), and the Fonds de recherche du Québec (FRQS, FRQSC and FRQNT). H.N. holds a Canada Research Chair funded by the Canadian Institutes of Health Research. Y.L. is supported by Natural Sciences and Engineering Research Council (NSERC) Discovery Grant (RGPIN-2019-0621), Fonds de recherche Nature et technologies (FRQNT) New Career (NC-268592), and Canada First Research Excellence Fund Healthy Brains for Healthy Life (HBHL) initiative New Investigator start-up award (G249591). The funders had no role in study design, data collection and analysis, decision to publish, or preparation of the manuscript.

**Competing interests:** The authors have declared that no competing interests exist.

summary statistics and functional annotations for accurate identification of causal variants. SparsePro extends the capabilities of a popular fine-mapping method, SuSiE, with important contributions in hyperparameter estimation, posterior summaries and integration of function annotations. Through extensive simulations, we demonstrate that our proposed approach can effectively integrate summary statistics and functional annotation, leading to improved power for identifying causal variants. Furthermore, we evaluate the benefits of incorporating functional annotations through real data analyses of five functional biomarkers. In summary, by improving power and providing valuable insights into complex disease genetics, SparsePro will have wide utility in advancing our knowledge and facilitating follow-up discoveries.

## Introduction

Establishment of large biobanks and advances in genotyping and sequencing technologies have enabled large-scale genome-wide association studies (GWAS) [1–3]. Although GWAS have revealed hundreds of thousands of associations between genetic variants and traits of interest, understanding the genetic architecture underlying these associations remains challenging [4–6], mainly because GWAS rely on univariate regression models that are not able to distinguish causal variants from other variants in linkage disequilibrium (LD) [5, 7, 8].

Several fine-mapping methods have been proposed to identify causal variants from GWAS. For instance, BIMBAM [9], CAVIAR [10] and CAVIARBF [11] estimate the posterior inclusion probabilities (PIP) for variants in a genomic locus by exhaustively evaluating likelihoods of all possible causal configurations. FINEMAP [12] accelerates such calculations with a stochastic shotgun search focusing on the most likely subset of causal configurations. However, the total number of variant-level causal configurations can grow exponentially with the number of causal variants, which can lead to a very high computational cost in classical fine-mapping methods. Starting from the motivation of quantifying uncertainty in selecting variants for constructing credible sets, SuSiE introduced a novel sum of single effect model and proposed an efficient iterative Bayesian stepwise selection (IBSS) algorithm [13, 14]. The IBSS algorithm sheds light to a promising approach to improve fine-mapping efficiency.

Additionally, functional annotations are commonly used as auxiliary evidence for prioritizing causal variants. PAINTOR [15] uses a probabilistic framework that integrates GWAS summary statistics with functional annotation data to improve accuracy of fine-mapping. Similarly, TORUS [16] incorporates highly informative genomic annotations to help with quantitative trait loci discoveries. Recently, PolyFun [17] was developed to use genome-wide heritability estimates from LD score regression to set the functional priors for fine-mapping methods. Given the computational efficiency of SuSiE, integrating functional annotations into similar algorithms can be desirable.

In this work, we present SparsePro for efficient fine-mapping with the ability to incorporate functional annotations. We connect the SuSiE IBSS algorithm with earlier work on a paired mean field variational inference algorithm [18] through the implementation of a sparse projection. We further propose effective strategies for estimating hyperparameters and summarizing posterior probabilities. We assess the performance of our proposed approach via simulation studies and examine the utility of SparsePro by fine-mapping five functional biomarkers of clinically relevant phenotypes.

## Results

### SparsePro method overview

To fine-map causal variants, SparsePro integrates two lines of evidence (Fig 1). First, with GWAS summary statistics and matched LD information, we jointly infer both the variant representations and the effect sizes for effect groups. Second, we estimate the functional enrichment of causal variants and use the estimates to further prioritize causal variants. As outputs, our method yields variant-level PIP estimates, set-level posterior summaries as well as enrichment estimates of functional annotations.

### Locus simulation

To evaluate the performance of SparsePro, we conducted simulations using the UK Biobank genotype data from multiple genomic loci (Methods), considering different numbers of causal variants and functional enrichment settings. As an example, the results from one locus (chr22: 31,000,000–32,000,000, Genome Assembly hg19) under a specific simulation setting: K = 5 (causal variants) and W = 2 (enrichment intensity) are shown in Fig 2. A reliable method should accurately identify true causal variants (represented by red dots) with high posterior inclusion probabilities (PIP) and assign low PIP to non-causal variants (represented by black dots), thereby achieving a high area under the precision-recall curve (AUPRC). Moreover, an ideal method should generate credible sets that have a high coverage and power while maintaining a small size.

We found that variant-level PIP obtained from FINEMAP, SuSiE, and SparsePro- (SparsePro implementation without functional information) were highly similar for most variants. Compared to FINEMAP, SparsePro- had fewer false positives; compared to SuSiE, SparsePro- was able to detect more causal variants with higher PIP (Fig 2A). Aggregating all simulation replicates, SparsePro- achieved an AUPRC of 0.91 in identifying true causal variants, while the second-best statistical fine-mapping method (SuSiE) achieved an AUPRC of 0.87 and PAINTOR- achieved an AUPRC of 0.82 (Fig 2B). Incorporating functional annotations can improve fine-mapping performance, with SparsePro+ achieving an AUPRC of 0.98, while PAINTOR + achieved an AUPRC of 0.93 (Fig 2B). Additionally, PIP generated by most methods exhibit good calibration, as the actual precision is close to the expected precision calculated by the mean PIP values (Fig 2C).

SparsePro yielded better set-level summaries compared to SuSiE due to its effective strategies for estimating hyperparameters and summarizing posterior probabilities (S1 Text). Overall, credible sets from SparsePro-, SparsePro+, and SuSiE exhibited similar coverage (Fig 3A). However, both SparsePro- and SparsePro+ consistently demonstrated a higher power and smaller size compared to SuSiE (Fig 3A). In simulation settings with functional enrichment, SparsePro+ demonstrated an additional increase in power and reduction in set size compared to SparsePro- (Fig 3A). For example, with K = 5 (causal variants) and W = 2 (enrichment intensity), SparsePro- and SuSiE achieved a similar mean coverage of 0.99 for their 95% credible sets (Fig 3A and S1 Table). However, SparsePro- outperformed SuSiE by achieving a higher mean power of 0.82 at a smaller mean size of 1.8, while SuSiE achieved a power of 0.68 at a mean size of 2.0 (Fig 3A and S1 Table). When functional information was incorporated, SparsePro+ further increased power and reduced the size of credible sets, achieving a power of 0.94 at a mean size of 1.4 (Fig 3A and S1 Table).

SparsePro also consistently demonstrated the highest AUPRC across all simulation settings (Fig 3B and S2 Table). When the number of causal variants was small (e.g. K = 1), the AUPRC of FINEMAP was comparable to SparsePro-; however, as the number of causal variants

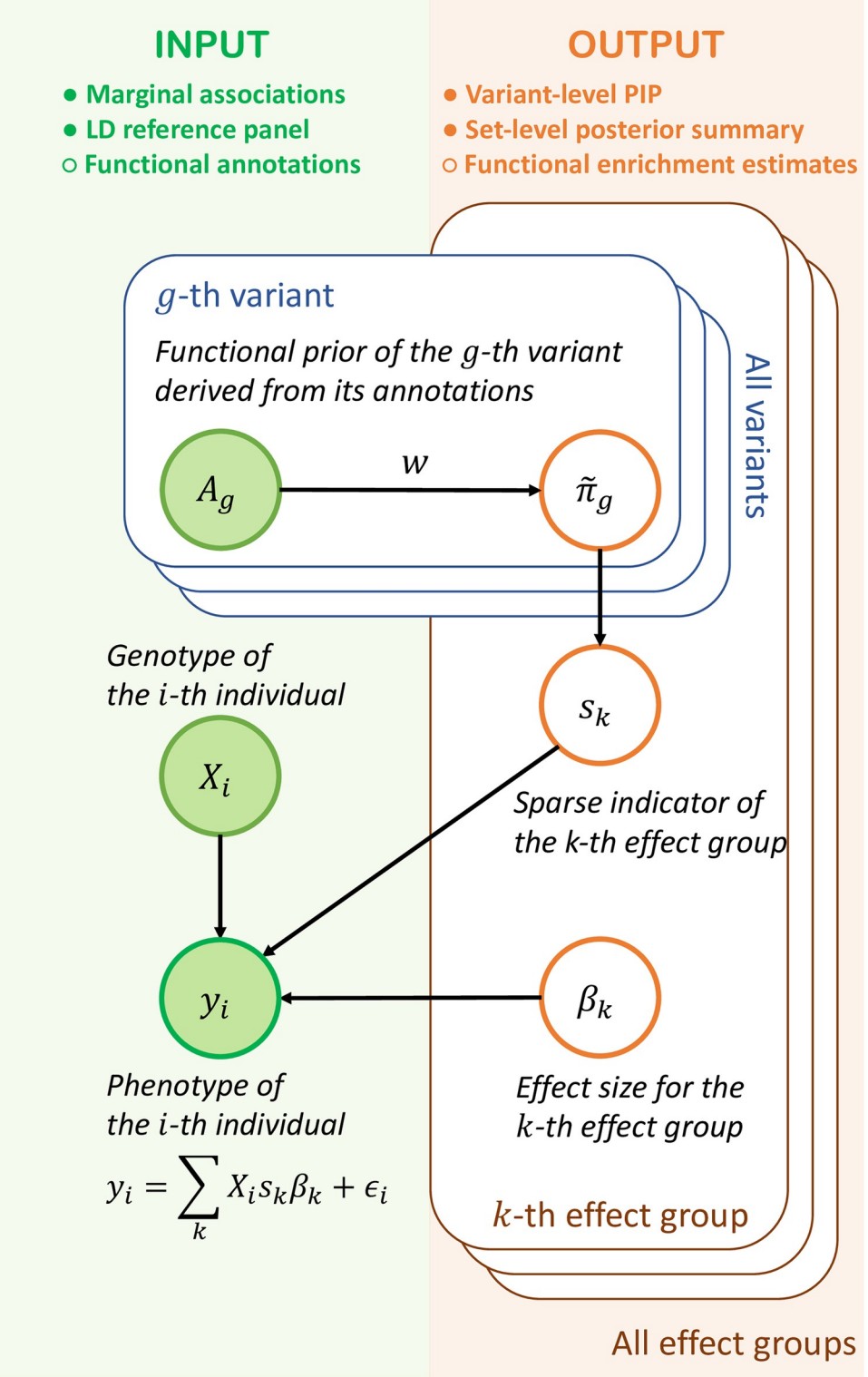

**Fig 1. SparsePro for integrating summary statistics and functional annotations.** The data generative process in SparsePro is depicted in this graphical model. Green shaded nodes represent observed variables: functional annotation information $A_g$ for the $g^{th}$ variant, genotype $X_i$ and trait $y_i$ for the $i^{th}$ individual. The orange unshaded nodes represent latent variables. Specifically, $\tilde{\pi}_g$ is the prior inclusion probability for the $g^{th}$ variant derived from functional annotation information; $\mathbf{s}_k$ is a sparse indicator specifying the variant representation of the $k^{th}$ effect group and $\beta_k$ represents the

effect size of the $k^{th}$ effect group. As a result, posterior summary can be obtained from posterior distribution of $\mathbf{s}_k$. Here, we assume individual-level data are available and adaption to GWAS summary statistics is detailed in the S1 Text.

increased, the performance of FINEMAP deteriorated (Fig 3B). Notably, without simulated functional enrichment (W = 0), none of the annotations demonstrated significance in the SparsePro G-test (Methods) (S3 Table). Therefore, SparsePro+ was not implemented. In contrast, PAINTOR did not select annotations by relevance, which could lead to worse performance of PAINTOR+ compared to PAINTOR- when there was no actual functional enrichment. For both PAINTOR+ and SparsePro+, stronger functional enrichment resulted in higher estimated enrichment weights (S3 and S4 Tables and S1 and S2 Figs). Consequently, the magnitude of their performance improvement became more evident compared to Sparse-Pro- and PAINTOR-, respectively(Fig 3B).

In these simulations, SparsePro also achieved great computational efficiency (Fig 3C), with the computation cost only increasing marginally as the number of causal variants increased. In contrast, while FINEMAP was the fastest approach when there was only one causal variant, its computational cost increased drastically with more causal variants (Fig 3C).

## Genome-wide simulation

In addition to locus simulations, we also performed genome-wide simulations to compare SparsePro with PolyFun, which relies on LD score regression to derive functionally-informed priors (Methods). A reliable functionally-informed fine-mapping method should accurately estimate the intensity of functional enrichment and incorporate this information into deriving functionally-informed priors. Additionally, the method should also demonstrate robustness against annotation misspecification or measurement errors.

SparsePro accurately estimated functional enrichment weights in different simulation settings. Specifically, when functional enrichment was simulated (W = 1 and W = 2), the Sparse-Pro G-test identified the enriched annotations as well as annotations that overlapped with the enriched annotations (S5 Table, S3 Fig). In settings without simulated functional enrichment (W = 0), none of the G-test results were statistically significant (S5 Table, S3 Fig). Furthermore, even without selecting relevant annotations based on a p-value threshold, the joint enrichment estimates in SparsePro yielded small weights for non-enriched annotations and large weights for enriched annotations, which closely aligned with their simulated values (S6 Table, S4 Fig). In comparison, the annotation coefficients obtained from LD score regression in PolyFun were small values that are difficult to interpret (S7 Table).

With accurate estimates of enrichment weights, SparsePro obtained well-informed priors that enhanced the power of causal variant prioritization. For instance, in the simulation setting with W = 2, SparsePro+ outperformed SparsePro- and other methods by identifying more causal variants with higher PIP (Fig 4A). The improved performance is demonstrated by achieving a higher AUPRC (S8 Table) and generating credible sets with an increased power and reduced size (S9 Table). The performance of SparsePro+PolyFun was worse than Sparse-Pro+, with a lower AUPRC (S8 Table) and credible sets with a reduced power (S9 Table).

The key distinction between the functionally-informed priors derived from SparsePro and PolyFun lay in their adaptability to the actual functional enrichment, as demonstrated through the logarithmic relative ratio (logRR) between the largest and smallest prior inclusion probabilities (Fig 4B). In SparsePro+, in settings with no simulated functional enrichment (W = 0), this ratio was 1 since a flat prior was used due to the absence of annotations selected by the G-

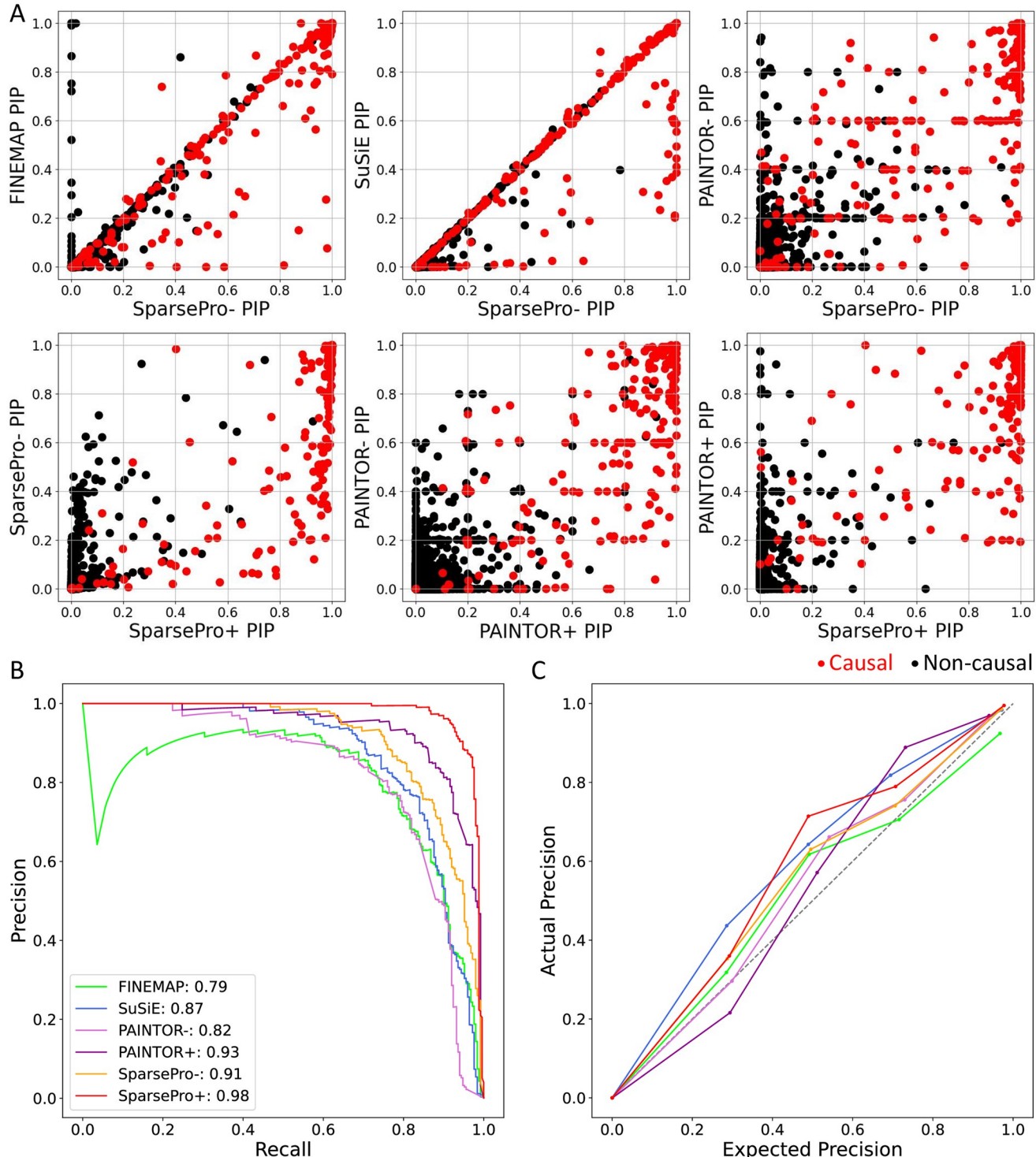

**Fig 2. Locus simulation in the setting of K = 5 (number of causal variants) and W = 2 (enrichment intensity).** (A) Comparison of posterior inclusion probabilities (PIP) obtained using different methods. Each dot represents a variant. True causal variants are colored red and non-causal variants are colored black. (B) Precision-recall curves. The area under the precision-recall curve (AUPRC) for each method is indicated. (C) Calibration curves. Variants are grouped into five bins according to their PIP values. Each dot represents one bin. The actual precision (y-axis) is plotted against the expected precision (x-axis) calculated by mean PIP values across all variants in the bin.

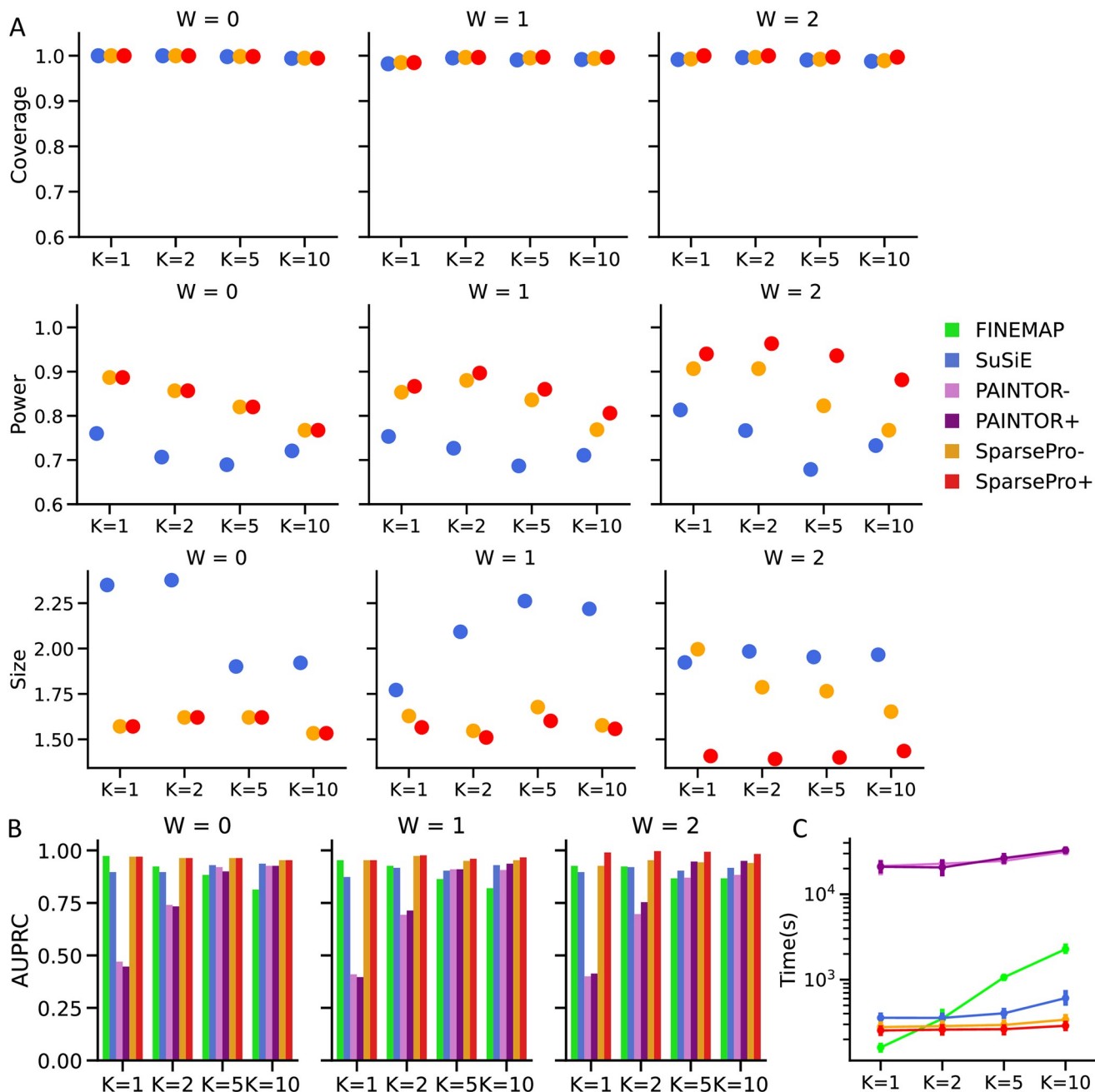

**Fig 3. Summary of locus simulations.** (A) Coverage, power and size of 95% credible sets. (B) Area under the precision-recall curve (AUPRC). (C) Computation time in seconds.

test for prior inclusion probability calculation (S10 Table, Fig 4B). As the enrichment intensity (W) increased, logRR also increased accordingly and were close to the simulated values (W = 1 or W = 2; S10 Table, Fig 4B). In contrast, in PolyFun, the logRR remained approximately log (100) as per the default setting of the algorithm, regardless of the actual functional enrichment (S10 Table, Fig 4B).

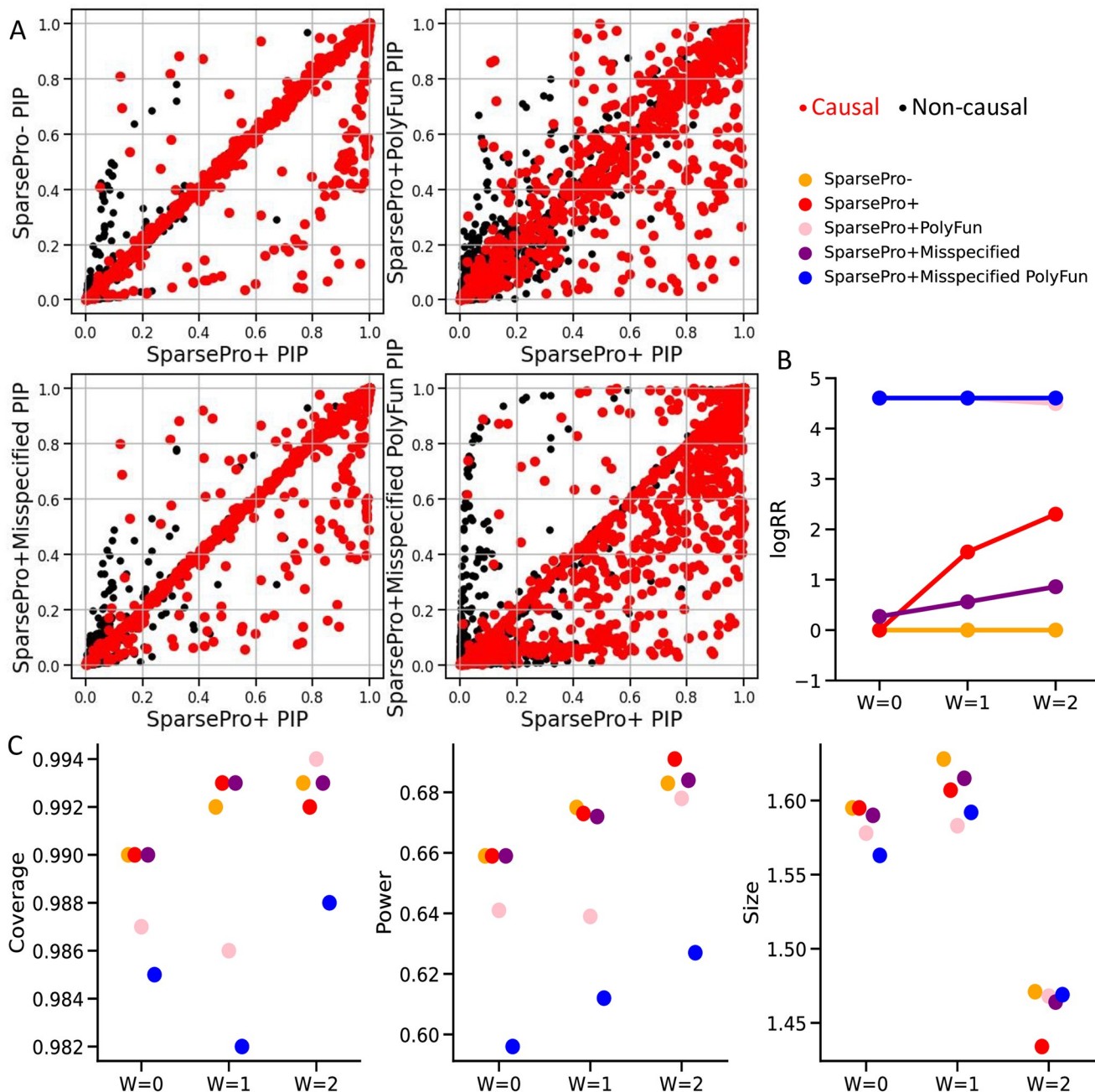

**Fig 4. Genome-wide simulations.** (A) Comparison of posterior inclusion probabilities (PIP) obtained using different methods in the simulation setting of W = 2. True causal variants are colored red and non-causal variants are colored black. (B) The logarithmic relative ratio (logRR) between the largest and smallest prior inclusion probabilities. (C) Coverage, power and size for 95% credible sets.

In the case of annotation misspecification (Methods), SparsePro demonstrated greater robustness compared to PolyFun. When the "conserved sequences" [19] annotation (the simulated misspecified annotation) was included in functional fine-mapping, the enrichment weights estimated by SparsePro+Misspecified were attenuated compared to SparsePro+ when the "non-synonymous" [20] annotation (the simulated enriched annotation) was included (Fig 4B). Consequently, SparsePro+Misspecified identified fewer causal variants with higher

PIP compared to SparsePro+ (Fig 4A, S5 and S6 Figs). However, SparsePro+Misspecified maintained a comparable performance to SparsePro- with a similar overall AUPRC (S8 Table) and similar coverage, power and size for credible sets (Fig 4C). In contrast, with the misspecified annotation, PolyFun still generated a strong functionally-informed prior (Fig 4B). As a result, a large number of non-causal variants were prioritized with high PIP (Fig 4A, S5 and S6 Figs), and the obtained credible sets had a lower coverage and reduced power (Fig 4C).

### Fine-mapping of functional biomarkers of clinically relevant phenotypes

We performed GWAS using data from the UK Biobank [1] for five functional biomarkers: FEV1-FVC ratio (FFR; lung function), estimated glomerular filtration rate (eGFR; kidney function), pulse rate (heart function), blood gamma-glutamyl transferase (gamma-GT; liver function) and blood glucose level (pancreatic islet function) (Methods). The fine-mapping results for these biomarkers demonstrated that functional annotations were informative in prioritizing causal variants. Notably, for all five biomarkers, the "non-synonymous" [20] annotation consistently exhibited the highest enrichment weights compared to other annotations (S7 Fig, S11 Table). Specifically, for eGFR, the non-synonymous annotation had an enrichment weight of 3.19 (95% CI: 2.88–3.50) (S11 Table). This indicates that non-synonymous variants were 24.3 (95% CI: 17.8–33.1) times more likely to be causal variants compared to variants that are not non-synonymous (S11 Table).

We used a different set of tissue-specific annotations to evaluate the biological relevance of the fine-mapping results (Methods). By estimating enrichment weights for these annotations based on variant-level PIP, we found that the tissue-specific annotations corresponding to each biomarker exhibited the highest enrichment weights (Fig 5A, S12 Table). For example, for eGFR, where the kidney-related annotation was the most relevant, the estimated enrichment weight from PIP derived from SparsePro- was 1.42 (95% CI: 1.25–1.59) while the estimated enrichment weight from PIP derived from SparsePro+ was 2.14 (95% CI: 1.95–2.33) (S12 Table).

Furthermore, the top variants from 95% credible sets also demonstrated tissue specificity (Fig 5B). For example, approximately 36% of the top variants for eGFR were annotated to kidney-specific annotations, while only 18% of the top variants for pulse rate were annotated to kidney-specific annotations (p-value from Fisher's exact test: $8.8 \times 10^{-7}$) (S13 and S14 Tables).

### Evidence of genetic coordination of clinically relevant phenotypes

The top variants from 95% credible sets mapped to genes in both core and regulatory pathways for these biomarkers (S15–S19 Tables). Four genes harbored top variants for four out of the five biomarkers (Fig 6A). Interestingly, we found that rs1260326 (Fig 6B), a missense variant (Leu446Pro) in gene *GCKR*, was fine-mapped for eGFR (PIP = 0.99), blood glucose level (PIP = 0.99), gamma-GT level (PIP = 1.00) and pulse rate (PIP = 0.85). Notably, this specific variant has been significantly associated with several glycemic traits [21] and quantitative traits for metabolic syndromes and comorbidities [22, 23], and has been implicated in the functions of the liver and other vital organs [24–26]. Other highly pleiotropic genes are transcription factors *GLIS3*, *RREB1* and *ZBT38*. These findings may present promising genetic targets for experimental validations in a larger effort towards understanding the mechanisms of genetic coordination among clinically relevant phenotypes.

## Discussion

Accurately identifying causal variants is fundamental to human genetics research and particularly important for interpreting GWAS results [5, 8]. In this work, we presented SparsePro to

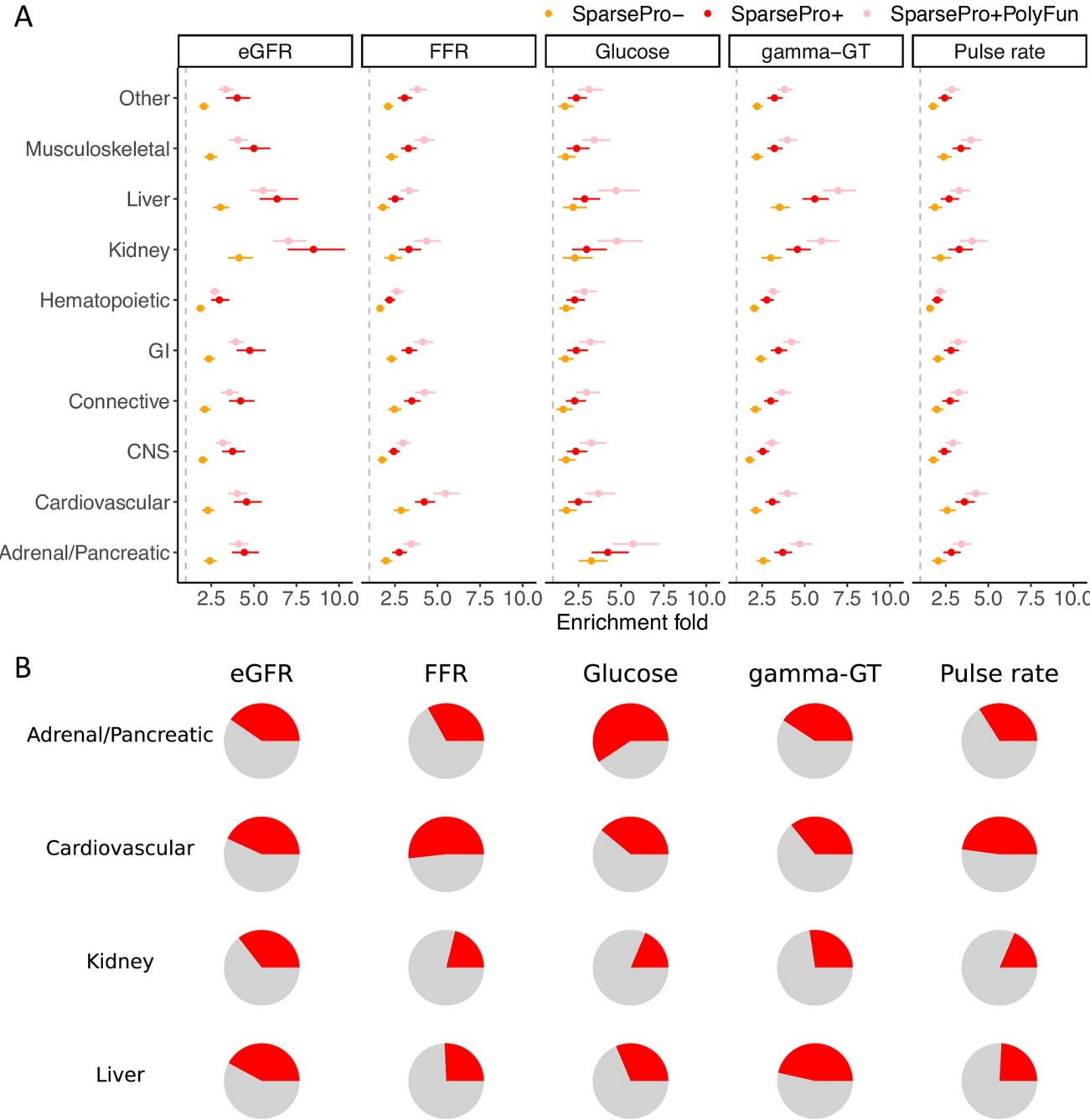

**Fig 5. Biological relevance of fine-mapping results for functional biomarkers of clinically relevant phenotypes.** (A) Enrichment fold in tissue-specific annotations. Each row denotes a tissue-specific annotation derived from histone marks (Methods) and each column denotes a functional biomarker. Error bars represent 95% confidence intervals for enrichment estimates. (B) Proportion of top variants from 95% credible sets mapped to tissue-specific annotations. Rows denote relevant tissue-specific annotations and columns denote functional biomarkers.

help prioritize causal variants by integrating GWAS summary statistics and functional information. We showcased the improved performance of our proposed approach through simulation studies. By fine-mapping genetic associations in five biomarkers of clinically relevant phenotypes, we demonstrated that functional annotations were useful in prioritizing biologically relevant variants.

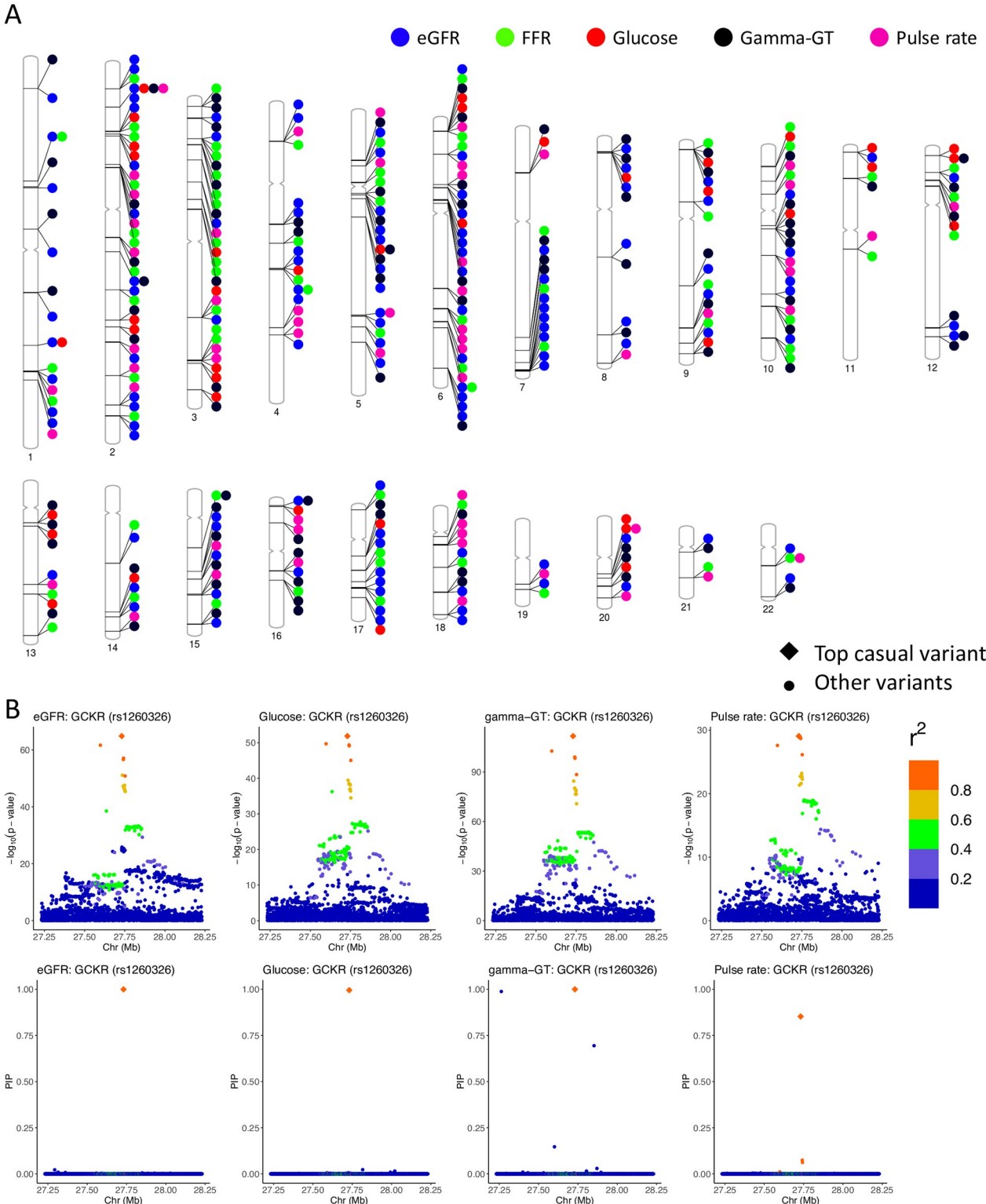

**Fig 6. Genes harboring causal sets for five functional biomarkers of clinically relevant phenotypes.** (A) Genome-wide distribution of genes harboring causal sets for at least two functional biomarkers. (B) *GCKR* locus with fine-mapped variant rs1260326. This variants was deemed causal for eGFR, glucose, gamma-GT and pulse rate. P-values from GWAS and posterior inclusion probabilities inferred from SparsePro+ are illustrated. Variants within a ±500kb window are colored by their linkage disequilibrium *r²* with rs1260326.

SparsePro builds upon SuSiE [13] and extends the capabilities of SuSiE with several important contributions. First, we proposed an effective strategy for estimating hyperparameters. Specifically, local heritability-based estimates can reduce the number of parameters to be estimated by the fine-mapping algorithm, resulting in improved power and efficiency. To showcase its utility, we also applied this strategy to SuSiE and observed substantial improvement of fine-mapping power (S2 Table) with calibrated PIP (S8 Table).

Moreover, we provided an alternative attainable coverage-based approach for posterior summaries. Specifically, we calculated attainable coverage for each effect group and only effect groups with attainable coverage greater than $\rho$ were summarized to $\rho$-level credible sets. We also applied this approach to SuSiE, which yielded improved set-level summaries compared to its original implementation with purity-based filtering (S9 Fig). As expected, if both strategies for estimating hyperparameters and summarizing posterior probabilities were incorporated in SuSiE, its performance could be comparable to that of SparsePro (S1 and S2 Tables).

Importantly, we provided a framework to integrate GWAS summary statistics and functional annotations. Functional annotations are widely used as additional evidence to prioritize causal variants together with statistical associations, with the possibility to elucidate the causal mechanisms [15–17]. In this study, we proposed an integrated approach for functional fine-mapping by jointly estimating enrichment weights for functional annotations and subsequently incorporating enrichment weights to derive functionally-informed priors. Therefore, the obtained priors were adaptive to functional enrichment based on the data, which allows the use of functional annotations in a cautious manner. We additionally introduced a G-test to assess the relevance of annotations. This G-test evaluates whether the causal signals are significantly enriched in the annotation of interest. In simulations, the G-test has shown its effectiveness in accurately identifying the enriched annotations (S3 and S5 Tables). However, in SparsePro, filtering annotations by the G-test does not impact the fine-mapping results dramatically (S2 and S8 Tables). This is because when irrelevant annotations are included in the joint estimates, their estimated enrichment weights are typically small (S6 Table), thus having a limited impact on the functionally-informed priors. Nonetheless, in SparsePro, screening annotations by G-test leads to simple interpretable models. While we used a p-value threshold of $1 \times 10^{-5}$ in both simulations and fine-mapping of functional biomarkers, users can adjust this threshold based on their preference for a more complicated model or a sparser model. Additionally, for other functionally-informed methods that are sensitive to annotation specifications, particularly those deriving strong priors from annotations, incorporating our proposed G-test can be useful to mitigate the impact of annotation misspecification.

In real data analyses, the "non-synonymous" [20] annotation is highly relevant in fine-mapping (S7 Fig) and indeed, by using this annotation to prioritize variants, we were able to identify rs1260326 as a causal variant for pulse rate when statistical evidence alone was not able to distinguish it from other variants in high LD (S10 Fig). However, future investigations are still needed to elucidate the roles of many other functional annotations.

Similar to existing fine-mapping algorithms, there are caveats in fine-mapping analysis using SparsePro. First, there are challenges related to allele flipping and LD rank deficiency when using summary statistics for fine-mapping. SparsePro, similar to Zou et al [14], does not require a full-rank LD matrix as it does not require matrix inversion throughout the algorithm. However, allele flipping can lead to algorithm convergence issues. To address this, it is recommended that users closely monitor the convergence of the algorithm and utilize scripts we provided to automate the formatting of GWAS summary statistics to match alleles in the LD reference panel. By taking these precautions, the potential convergence issues caused by allele flipping can be mitigated. Additionally, the identification of causal variants in fine-mapping

relies on the rigorousness of GWAS study design, and may be biased if unmeasured confounding factors such as population stratification are not properly controlled for.

In summary, SparsePro is an accurate and efficient fine-mapping method integrating statistical evidence and functional annotations. We envision its wide utility in understanding the genetic architecture of complex traits, identifying target genes, and increasing the yield of functional follow-up studies of GWAS.

## Methods

### SparsePro for efficient fine-mapping integrating summary statistics and functional annotations

In SparsePro, we use a generative model to integrate GWAS summary statistics and functional annotations (Fig 1 and S1 Text). First, we specify prior inclusion probability for the $g^{th}$ variant $\tilde{\pi}_g$:

$$\tilde{\pi}_g = \frac{\exp(\mathbf{A}_g^T \mathbf{w})}{\sum_{g'=1}^{G} \exp(\mathbf{A}_{g'}^T \mathbf{w})}$$

where $\mathbf{A}_g$ is the $M \times 1$ vector of $M$ annotations for the $g^{th}$ variant and $\mathbf{w}$ is a $M \times 1$ vector of enrichment weights. Here, we use the *softmax* function to ensure the prior probabilities are normalized. If no functional information is provided, the prior inclusion probability is considered equal for all variants.

Subsequently, we assume the high dimensional genotype matrix $\mathbf{X}_{N \times G}$ can be represented by altogether $K$ effect groups via a sparse projection $\mathbf{S}_{G \times K} = [\mathbf{s}_1, \ldots, \mathbf{s}_K]$ with

$$\mathbf{s}_k \sim \mathbf{Multinomial}(1, \tilde{\pi})$$

Then the effect sizes for effect groups can be represented by $\boldsymbol{\beta} = [\beta_1, \ldots, \beta_K]$ where

$$\beta_k \sim \mathcal{N}(0, \tau_\beta^{-1})$$

Finally, for a continuous trait $\mathbf{y}_{N \times 1}$ over $N$ individuals, we have:

$$\mathbf{y} \sim \mathcal{N}(\mathbf{X}\mathbf{S}\boldsymbol{\beta}, \tau_y^{-1}\mathbf{I})$$

For inference, we use an efficient paired mean field variational inference algorithm [18] adapted for GWAS summary statistics, which we show is equivalent to the SuSiE IBSS algorithm [13] (detailed in the S1 Text). We estimate hyperparameters for effect sizes $\tau_\beta$ and residual errors $\tau_y$ using local heritability-based estimates from HESS [27] (S1 Text) and propose an attainable coverage-based strategy for summarizing posterior probabilities (S1 Text). Additionally, we use joint estimates of enrichment weights to derive functionally-informed priors to further prioritize causal variants (S1 Text) and introduce a G-test to screen relevant functional annotations (S1 Text).

### Locus simulation studies

We conducted locus simulations to evaluate the performance of fine-mapping methods under different settings. We randomly selected three 1-Mb regions, and obtained genotypes for 353,570 unrelated UK Biobank White British ancestry individuals [1]. For each locus, we generated 50 replicates for each combination of parameters: $K \in \{1, 2, 5, 10\}$ (number of causal variants) and $W \in \{0, 1, 2\}$ (enrichment intensity) among variants that were annotated as "conserved sequences" [19], "DNase I hypersensitive sites" (DHS) [28], "non-synonymous" [20], or

overlapping with histone marks H3K27ac [29] or H3K4me3 [28]. In the simulated weight vector **w**, the entries that correspond to the these enriched annotations had a value of $W$. Causal variants in each simulation replicate were randomly assigned. Then, we used the GCTA GWAS simulation pipeline [30] to simulate a continuous trait with a total heritability of $K \times 10^{-4}$. We performed association test between each variant and the simulated trait, and obtained GWAS summary statistics using the fastGWA software [31].

Next, we ran the different fine-mapping programs with the GWAS summary statistics and in-sample LD as inputs. For methods using functional annotations, we provided the aforementioned five annotations with enrichment of causal variants as well as five additional annotations without enrichment: "actively transcribed regions" [32], "transcription start sites" [32], "promoter regions" [33], "5'-untranslated regions" [20], and "3'-untranslated regions" [20]. The statistical fine-mapping results obtained from SparsePro without annotation information were denoted as "SparsePro-". Annotations with a G-test p-value $< 1 \times 10^{-5}$ were selected for functionally-informed fine-mapping, and the results were referred to as "SparsePro+". Additionally, we performed functionally-informed fine-mapping by including all annotations (i.e., a G-test p-value $< 1.0$) without G-test screening, denoted as "SparsePro+1.0". Moreover, we conducted statistical fine-mapping using the stochastic shotgun search mode of FINEMAP (V1.4) and the function "susie_rss" from SuSiE (V0.12.16). The mcmc mode for PAINTOR (V3.0) was used to obtain the baseline model results and the annotated model results, separately denoted as "PAINTOR-" and "PAINTOR+". The largest $K$ used for SparsePro, SuSiE and FINEMAP was 10. Due to the high computation cost, PAINTOR only allows up to 3 causal variants per locus. Computation time was recorded on a 2.1 GHz CPU for fine-mapping programs including all procedures.

Furthermore, we investigated the benefits of our proposed strategies for estimating hyperparamters and summarizing posterior probabilities (detailed in the S1 Text) by incorporating them into SuSiE. Specifically, local heritability-based estimates for effect size variance and residual variance were provided to "scaled_prior_variance" and "residual_variance" respectively in both SuSiE+HESS and SuSiE+SparsePro while the default empirical Bayes based hyperparameter estimates were used in SuSiE. The posterior summaries obtained from SuSiE with heritability-based hyperparameters were denoted as "SuSiE+HESS" while the posterior summaries obtained using our proposed approach were denoted as "SuSiE+SparsePro" (S1 Text).

## Genome-wide simulation studies

We conducted genome-wide simulations to compare SparsePro+ with other methods that requires genome-wide GWAS summary statistics for functional fine-mapping. We obtained genotypes of 353,570 unrelated UK Biobank White British individuals on chromosome 22 and sampled 100 causal variants with $W \in \{0, 1, 2\}$ (enrichment intensity) among variants that were annotated as "non-synonymous" [20]. We used the GCTA GWAS simulation pipeline [30] to simulate a continuous trait with a per-chromosome heritability of 0.01. We tested the association between each variant and the simulated trait, and obtained GWAS summary statistics using the fastGWA software [31]. This process was repeated 22 times to obtain genome-wide GWAS summary statistics. Additionally, we obtained LD information calculated using the UK Biobank participants from Weissbrod et al [17]. These LD matrices were generated for genome-wide variants binned into sliding windows of 3 Mb with neighboring windows having a 2-Mb overlap.

We applied SparsePro to the GWAS summary statistics with the aforementioned LD information, iterating over all sliding windows initially without any functional annotation. The

fine-mapping results obtained were referred to as "SparsePro-". Next, the 10 annotations used in locus simulations were used to derive functional priors. The fine-mapping results from SparsePro with a prior derived from PolyFun were denoted as "SparsePro+PolyFun". Additionally, results from SparsePro with a functional prior estimated from annotations with a G-test p-value less than $1 \times 10^{-5}$ were denoted as "SparsePro+" while results from SparsePro with a functional prior estimated from all 10 annotations were denoted as "SparsePro+1.0". In these fine-mapping analyses, variants in each 3-Mb sliding window were fine-mapped jointly. However, we only retained PIP for variants located in the 1-Mb region central to the window as well as credible sets with top variants located in this 1-Mb region. Therefore, variants were fine-mapped together with neighboring variants within at least 1-Mb to mitigate boundary effect.

To further investigate the impact of annotation misspecification or annotation measurement errors on functionally-informed fine-mapping, we utilized the "conserved sequences" [19] annotation, which partly overlaps with the simulated enriched "non-synonymous" [20] annotation. We used this annotation for deriving the functional prior using both SparsePro and PolyFun, and the corresponding results were labeled as "SparsePro+Misspecified" and "SparsePro+Misspecified PolyFun", respectively. These analyses allowed us to evaluate the robustness of the functionally-informed fine-mapping approach to annotation misspecifications and potential measurement errors.

## Fine-mapping of functional biomarkers of clinically relevant phenotypes

To investigate potential genetic coordination mechanisms, we performed GWAS in the UK Biobank [1], focusing on five functional biomarkers: forced expiratory volume in one second to forced vital capacity (FEV1-FVC) ratio for lung function, estimated glomerular filtration rate for kidney function, pulse rate for heart function, gamma-GT for liver function and blood glucose level for pancreatic islet function. For each biomarker, we first regressed out the effects of age, $age^2$, sex, genotyping array, recruitment centre, and the first 20 genetic principal components before inverse normal transforming the residuals to z-scores that had a zero mean and unit variance. We then performed GWAS analysis on the resulting z-scores with the fastGWA software [30, 31] to obtain summary statistics.

Using the summary statistics and the matched LD information [17], we performed genome-wide fine-mapping with "SparsePro-", "SparsePro+" and "SparsePro+PolyFun" as described in Section 5.3 with annotations from the "baselineLF2.2.UKB" model [17] provided by PolyFun.

To assess the biological relevance of fine-mapping results, we used 10 tissue-specific annotations derived from four histone marks H3K4me1, H3K4me3, H3K9ac, and H3K27ac by Finucane et al [34]. This set of annotations was not used by any functional fine-mapping methods. To assess tissue specificity of the obtained PIP values, we ran G-test and estimated enrichment weight (S1 Text) for each tissue-specific annotation. Additionally, we examined whether the top variants from 95% credible sets identified for a trait were more enriched for relevant tissue-specific annotations compared to the top variants identified for other traits by Fisher's exact tests.

We used phenogram [35] to illustrate genes that harbored causal variants for at least two biomarkers to explore possible pleiotropic effects.

## Supporting information

**S1 Text. Supplementary notes.**
(PDF)

**S1 Table. Summary of coverage, power and size of 95% credible sets in locus simulations.**
(XLSX)

**S2 Table. Summary of AUPRC in locus simulations.**
(XLSX)

**S3 Table. Annotation enrichment weights and G-test p-values from SparsePro in locus simulations.**
(XLSX)

**S4 Table. Annotation enrichment weights estimated by PAINTOR+ in locus simulations.**
(XLSX)

**S5 Table. Annotation enrichment weights and G-test p-values from SparsePro in genome-wide simulations.**
(XLSX)

**S6 Table. Annotation enrichment weights estimated jointly from SparsePro in genome-wide simulations.**
(XLSX)

**S7 Table. Annotation coefficients estimated by PolyFun in genome-wide simulations.**
(XLSX)

**S8 Table. Summary of AUPRC in genome-wide simulations.**
(XLSX)

**S9 Table. Summary of coverage, power and size of 95% credible sets in genome-wide simulations.**
(XLSX)

**S10 Table. Summary of the relative ratio between the largest and smallest prior inclusion probabilities in genome-wide simulations.**
(XLSX)

**S11 Table. Annotation enrichment weights and G-test p-values from SparsePro in fine-mapping functional biomarkers.**
(XLSX)

**S12 Table. Annotation enrichment weights for tissue-specific annotations and G-test p-values from SparsePro-, SparsePro+ and SparsePro+PolyFun in fine-mapping functional biomarkers.**
(XLSX)

**S13 Table. Percentage of top variants from 95% credible sets annotated to tissue-specific annotations in fine-mapping functional biomarkers.**
(XLSX)

**S14 Table. Fisher's exact for tissue specificity in fine-mapping functional biomarkers.**
(XLSX)

**S15 Table. 95% credible sets for eGFR from SparsePro+.**
(XLSX)

**S16 Table. 95% credible sets for FFR from SparsePro+.**
(XLSX)

**S17 Table. 95% credible sets for gamma-GT from SparsePro+.**
(XLSX)

**S18 Table. 95% credible sets for glucose from SparsePro+.**
(XLSX)

**S19 Table. 95% credible sets for pulse rate from SparsePro+.**
(XLSX)

**S1 Fig. Annotation enrichment weights estimated by SparsePro in locus simulations.** Each grid corresponds to a different simulation setting of K (number of causal variants) and W (enrichment intensity). Error bars represent 95% confidence intervals for enrichment estimates. Blue dots are estimated values and yellow triangles are simulated values.
(TIFF)

**S2 Fig. Annotation enrichment weights estimated by PAINTOR in locus simulations.** Each grid corresponds to a different simulation setting of K (number of causal variants) and W (enrichment intensity). PAINTOR does not provide confidence intervals for enrichment weights. Blue dots are estimated values and yellow triangles are simulated values.
(TIFF)

**S3 Fig. Annotation enrichment weights estimated by SparsePro in genome-wide simulations.** Each row represents a different simulation setting with W (enrichment intensity) = 0, 1, or 2. Error bars represent 95% confidence intervals for enrichment estimates. Blue dots are estimated values and yellow triangles are simulated values.
(TIFF)

**S4 Fig. Enrichment weights estimated jointly by SparsePro without filtering annotations in genome-wide simulations.** Each row represents a different simulation setting with W (enrichment intensity) = 0, 1, or 2. Error bars represent 95% confidence intervals for enrichment estimates. Blue dots are estimated values and yellow dots are simulated values.
(TIFF)

**S5 Fig. Comparison of posterior inclusion probabilities (PIP) obtained using different methods in the simulation setting of W (enrichment intensity) = 1.** True causal variants are colored red and non-causal variants are colored black.
(TIFF)

**S6 Fig. Comparison of posterior inclusion probabilities (PIP) obtained using different methods in the simulation setting of W (enrichment intensity) = 0.** True causal variants are colored red and non-causal variants are colored black.
(TIFF)

**S7 Fig. Enrichment fold of annotations in fine-mapping functional biomarkers.** Each row denotes an annotation and each column denotes a functional biomarker. Error bars represent 95% confidence intervals for enrichment estimates.
(TIFF)

**S8 Fig. Calibration curves for SuSiE and SuSiE with hyperparameters estimated from HESS.** Variants are grouped into five bins according to their PIP values. Each dot represents one bin. The actual precision (y-axis) is plotted against the expected precision (x-axis) calculated by mean PIP values across all variants in the bin.
(TIFF)

**S9 Fig. Summary of coverage, power and size of 95% credible sets in different simulation settings for SuSiE, SuSiE with hyperparameters estimated from HESS (SuSiE+HESS) and SuSiE with hyperparameters estimated from HESS and posterior summaries proposed in SparsePro (SuSiE+SparsePro).**
(TIFF)

**S10 Fig. Fine-mapping the GCKR locus for pulse rate.** (A) GWAS summary statistics for pulse rate at the GCKR locus. (B) Fine-mapping results from SparsePro-. (C) Fine-mapping results from SparsePro+. (D) Fine-mapping results from SparsePro+PolyFun. P-values from GWAS and inferred posterior inclusion probabilities from fine-mapping are illustrated. Variants within a ±500kb window are colored by their linkage disequilibrium $r^2$ with rs1260326.
(TIFF)

**S11 Fig. Comparison of posterior inclusion probabilities (PIP) in the simulation setting of K (number of causal variants) = 5 and W (enrichment intensity) = 2 between SuSiE and SuSiE with hyperparameters estimated from HESS (SuSiE+HESS).** True causal variants are colored red and non-causal variants are colored black.
(TIFF)

**S12 Fig. Comparison of posterior inclusion probabilities (PIP) in the simulation setting of K (number of causal variants) = 5 and W (enrichment intensity) = 2 between SparsePro-(SparsePro without functional annotations) and SuSiE with hyperparameters estimated from HESS (SuSiE+HESS).** True causal variants are colored red and non-causal variants are colored black.
(TIFF)

**S13 Fig. Comparison of posterior inclusion probabilities (PIP) in the simulation setting of K (number of causal variants) = 5 and W (enrichment intensity) = 2 between SuSiE with hyperparameters estimated from HESS (SuSiE+HESS) and SuSiE with hyperparameters estimated from HESS and posterior summaries proposed in SparsePro (SuSiE+SparsePro).** True causal variants are colored red and non-causal variants are colored black.
(TIFF)

## Acknowledgments

This study has been conducted using UK Biobank Resources under Application Number 45551 and we thank NeuroHub for providing access to data resources. This study was enabled, in part, by support from Calcul Québec and Compute Canada. We thank Dr. Robert Sladek and Dr. Josée Dupuis for helpful discussion and suggestions.

## Author Contributions

**Conceptualization:** Wenmin Zhang, Yue Li.

**Data curation:** Wenmin Zhang.

**Formal analysis:** Wenmin Zhang.

**Funding acquisition:** Wenmin Zhang, Hamed Najafabadi, Yue Li.

**Investigation:** Wenmin Zhang.

**Methodology:** Wenmin Zhang.

**Project administration:** Wenmin Zhang, Hamed Najafabadi, Yue Li.

**Resources:** Wenmin Zhang, Hamed Najafabadi, Yue Li.

**Software:** Wenmin Zhang.

**Supervision:** Hamed Najafabadi, Yue Li.

**Validation:** Wenmin Zhang, Hamed Najafabadi, Yue Li.

**Visualization:** Wenmin Zhang.

**Writing – original draft:** Wenmin Zhang.

**Writing – review & editing:** Wenmin Zhang, Hamed Najafabadi, Yue Li.

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
