## [Decision Letter · Decision Letter 0]

15 Mar 2023

Dear Dr Zhang,

Thank you very much for submitting your Methods entitled 'SparsePro: an efficient fine-mapping method integrating summary statistics and functional annotations' to PLOS Genetics.

The manuscript was fully evaluated at the editorial level and by independent peer reviewers. The reviewers appreciated the attention to an important problem, but raised some substantial concerns about the current manuscript. The main concern raised by all reviewers is the strong similarity between the proposed variational inference framework and the SuSiE model (Wang et al., 2020). Reviewers are unsure whether the subtle changes made to SuSiE in the proposed framework are a misinterpretation of the original model or have well-justified reasons. To address this concern, the editors suggest submitting a new manuscript that uses the current work as a starting point. This new manuscript should explicitly connect the proposed framework to the SuSiE model and algorithm, explain the motivation for modifications made to SuSiE before incorporating annotations, and provide comments on why these changes are necessary. Based on the reviews, we will not be able to accept this version of the manuscript, but we would be willing to review a much-revised version. We therefore suggest major revision with these important details clarified. We cannot, of course, promise publication at that time. 

If you decide to revise the manuscript for further consideration at PLOS Genetics, please aim to resubmit within the next 60 days, unless it will take extra time to address the concerns of the reviewers, in which case we would appreciate an expected resubmission date by email to plosgenetics@plos.org.

We are sorry that we cannot be more positive about your manuscript at this stage. Please do not hesitate to contact us if you have any concerns or questions.

Yours sincerely,

Gao Wang

Guest Editor

PLOS Genetics

Xiaofeng Zhu

Section Editor

PLOS Genetics

Reviewer's Responses to Questions

**Comments to the Authors:**

Reviewer #1: Zhang et al present their extension of the SuSiE model to incorporate functional annotations. SparsePro prevents irrelevant annotations from contaminating the model by testing the functional annotations before integrating them into the model. It estimates the enrichment coefficients within the IBSS algorithm. Moreover, it estimates some hyper-parameters outside the IBSS algorithm to avoid convergence issue. The manuscript presents simulation results to support the method. They also show application result on UKBiobank traits. It is well written. As SuSiE becomes a popular approach for fine-mapping, it is desirable to have a version supporting functional annotations.

Major comments:

1. The SparsePro model is essentially the SuSiE model with functional annotations, with some changes to the prior and residual variance estimation. As most readers are familiar with SuSiE, I suggest using the terminology and concepts from SuSiE in the manuscript. For example, replacing 'effect group' with 'credible set'; using similar credible set definition as in SuSiE. The derivations in supplementary section 1 before annotation estimation are same as in SuSiE, but the SuSiE paper is not cited.

2. The statement on line 214 said 'in SuSiE, Bayes factors were normalized by sum of Bayes factors … which increased power for identifying causal variants.' This statement is not accurate. The softmax is same as normalizing weighted Bayes Factors (weighted by the prior probability \\pi_g). So the log probability in page 37 3rd equation is same as log(\\pi_g * Bayes Factor). This is not the cause for the SparsePro higher power in the simulation. The higher power from SparsePro- may caused by estimating hyper-parameters outside the iterative algorithm, and different PIP definition.

3. The PIP for each variant is computed as max among groups. Why not computing using 1-\\prod_k(1-\\gamma_kg), which is the theoretical definition of PIP?

4. From line 310-313, it seems that the 95% causal set (1 set containing all causal variants) is used in the set-level comparisons, not 95% credible sets as defined in SuSiE. Is there any reason not using credible sets for comparisons? SparsePro, SuSiE and FINEMAP all output 95% credible sets. It is unnecessary to combine them into one causal set. I suggest conducting set-level comparisons using 95% credible sets (coverage / power / size of each single 95% credible set).

5. The method estimates the enrichment coefficient using 'one-at-a-time' coordinate ascent. Is this same as jointly estimating the coefficients? Is it possible to estimate them jointly? For your reference, TORUS (Wen, X., 2016.) estimates the coefficients jointly using EM.

Minor comments:

1. The credible sets from SuSiE has a purity > 0.5 filter by default. Is the same purity filter applied in other methods? If not, the comparison is unfair.

2. The simulation has a per-variant heritability 10^(-4). Does this mean '--simu-hsq' is K * 10^(-4) in GCTA simulation? Or does it mean the simulated causal variants have similar effect size?

3. The relationship between the simulation parameter W and the relative enrichment vector w in section 4.1 page 10 is unclear.

4. Line 242 'causal effect sizes may vary across different subpopulations', where does the subpopulation come from? This is just single study fine-mapping.

5. Do the computational times in figures include estimating \\tao_\\beta, tao_y and testing of functional annotations?

6. Any reason to use log20 for entropy difference cutoff? Any reason to use 10^(-5) p value threshold for G-test?

7. In the genome-wide simulation, the results in the 1-MB center of each 3-MB window are considered. But the signal could also at/close to the 1-MB center boundary. How to analyze these signals?

8. What's the coefficient w scale in Fig S7?

Reviewer #2: Comment,

Zhang et al. proposed an interesting enhancement of the SuSiE model proposed by Wang et al. JRSSB 2021 to perform informed fine-mapping using side information/annotation. The authors show some convincing evidence that their approach (SparsePro) has greater power than some competing methods, such as SuSiE+Polyfun. Overall the approach is sound, and I am mostly positive regarding the manuscript. The level of detail provided by the authors is satisfactory. Still, some typos are disrupting the overall quality of the manuscript as well as some statements that need to be clarified.

Major comment:

1) While most of the manuscript reads quite well, there is a couple of sentences that do not flow well or are grammatically incorrect, e.g., in the last paragraph of the introduction, the authors wrote

"In line with the idea of grouping correlated variants together into effect groups, we proposed Sparse Projections to Causal Effects (SparsePro) to further improve fine-mapping efficiency and accuracy. First, within each effect group, we additionally incorporate" in this case, first needs to be followed by a sentence with an active verb before additionally. I am not a native English speaker, and I am aware that it can be hard to draft a manuscript. However, before considering the manuscript for acceptance, the authors need to proofread the manuscript to correct some of these problematic sentences.

2) The authors wrote: " Second, we use an efficient variational

inference algorithm to further simplify the intuitive algorithm proposed in SuSiE and improve computation efficiency." I have had a close look at the algorithm proposed by Zhang. While the author does not explicitly compute the marginal Bayes factor for each variant, given that we do not use any annotation, the coordinate ascend seems very similar to the one proposed by Wang et al.

As Wang and colleague provided the complexity of each coordinate ascend update, I think it would be interesting that the authors provide the complexity of each coordinate ascend update of SparsePro. This would make it more explicit that the gain in computational speed is not only due to the implementation. Because for the moment, it is not clear to me how this approach differs from the IBSS. Perhaps the author could elaborate on the computational complexity of their VA of the Single Effect Model proposed by Wang et al.

3) I would be interested in seeing a set of simulations in which the annotations are misspecified/measured with noise and potential bias toward non-causal SNPs. While substantial efforts are made to get high-quality annotations, it is not unlikely that many of those are poorly measured or biased. I would be interested in seeing some simulations in which the author would consider poorly measured annotations and see if that could generate some low-coverage credible set. In general, my overall question is that given that you have some annotations, is it worth it to include them in a fine mapping procedure, or could that potentially "harm" your results. Could you generate a new set of simulations in which annotations are measured with noise? Furthermore, could try to come up with a set of simulations that could lead to problematic coverage due to annotations. For example, suppose that you use the following annotation (that is made to be problematic) in a case where you consider a model with K SNP. Consider the following K annotation for each non-casual SNP set annotation k to its correlation with causal SNP k and for the causal SNP set each of their K annotations by sampling a random number 0 and 1

Minor:

1) There is a type after equation 2 in the supplement for the condition variational approximation. It is written $s_{kg=1}$ whereas it should be $s_{kg}=1$; please go through the equation to correct the other typos

2) The equation below, "Therefore, the posterior probability of the gth variant being causal in the kth effect group can be estimated as:" seems somewhat not correct. The input of in the softmax function is a scalar, whereas it should be a vector. The posterior probability of the gth variant being causal in the kth effect group should be the g component of this softmax.

3) In the SuSiE-rss manuscript, Zou and colleagues spend a substantial amount of work dealing with problematic LD. I would be interested to explicitly say what is implemented in SparsePro to circumvent problems related to LD matrices that are not full rank or allele flipping problem

4) could you show if the Gtest used for testing the annotation is correctly controlling the type I error

Reviewer #3: The authors propose a method for estimating the hyperparameters of the sum of single effects regression. In particular they leverage functional annotation enrichments to specify informative prior inclusion probabilities for different variants, and leverage heritability estimates to set the effect size and residual variance hyper-parameters. By incorporating this enrichment information they are able to demonstrate improvement over fine-mapping methods that either do not leverage functional annotations, or leverage functional annotations through different means (e.g. Polyfun, which computes prior inculsion probabilities given partitioned heritability estimates across a set of annotations). These are important contributions, as the selection of these hyperparameters can greatly influence the calibration and power of finemapping.

While I am generally positive about the work put forth here, my main reccomendations are to focus the discussion and commentary on the benefits of including functional annotations, and providing more clear rationale for the use of heritability information to set the effect and error variance hyperparameters. The authors should modify their discussion of the algorithmic differences between SuSiE and SparsePro because they do not seem accurate-- as far as I can tell, for a fixed set of hyperparameters (prior inclusion probability, effect variance, residual variance) the coordinate ascent variational inference (CAVI) employed for SparsePro and SuSiE's IBSS algorithm (which is also CAVI) are the same.

**Estimating prior inclusion probabilities** (Sparspro vs Polyfun) To my understand, Polyfun provides a heuristic for forming the prior inclusion probabilites based on a heritability partition. Sparsepro takes a less heuristic approach by directly estimating enrichment of selected variants, and using those enrichments to refine the posterior approximations made by SuSiE. I very much support this approach, and the authors successfully show through extensive simulations how their method improves on the heursitic approach to estimating the prior inclusion probabilities developed in Polyfun.

**Estimating variance hyperparametes** (Sparsepro vs SuSiE) SuSiE uses a variational empricial Bayes approach to estimate the effect variance and residual variance-- this just means optimizing the objective w.r.t to these hyperparameters. In contrast, Sparsepro fixes these hyperparemeters to values informed by heritability estimates. For example, the residual variance is set to 1-h2.

The residual variance is fixed to 1 - h2 where h2 is a locus level heritability estimate. In contrast, a conservative approach would be to set the residual variance to 1. I'm concerned that setting the residual variance to 1-h2 may disrupt calibration of the posterior. Basically, while h2 is an estimate of the heritability in the locus, finemapped association signals will only explain a portion of this heritability. 1-h2 may be too small, and encourage the model to select variants in a way that is anti-conservative.

**The variational approximations are identical** The discussion and supplemental materials emphasize the differences in computation between SuSiE's IBSS algorithm, and the variational updates derived in this paper. However, it is important to note that the variational approximation for Sparsepro and SuSiE are identical $q(\\beta, S) = \\prod_k q(\\beta_k, s_k)$. Consequently all differences in performance between SuSiE and Sparsepro- (without annotation) should be explained by (1) differences in the hyperparameters/hyperparameter estimation procedure and (2) implimentation details (e.g. convergence criteria, order of coordinate updates, etc., which may influence which local optima of the variational objective is found).

In particular the following does not seem correct 214:215 "In SuSiE, Bayes factors were normalized by sum of Bayes factors across all variants while SparsePro uses the softmax function to normalize posterior probabilities which increased power for identifying causal variants". I believe the marginal log Bayes factors are equal (up to a constant) to the posterior (log) probabilities referenced here (th 4th expression in supplementary material, page 27). Thus normalizing Bayes factors is equivalent to applying softmax of the log probabilities.

**Suggested Revisions**

- Clarify the similarities and differences between Sparsepro and SuSiE. I believe the variational approximations are the same, but the real contribution here are annotation and heritability informed hyperparameter settings, which are an important contribution that can stand on there own.

- Sparsepro- and SuSiE shoud be identical up the the setting of the effect variance and residual variance hyperparameters. Commentary attempting to explain the difference in performance between SuSiE and Sparsepo- should be revised, because at times it implies a difference in the algorithm/optimization procedure which does not seem to be correct.

- Please discuss/justify the heritability based estimates for effect variance and residual variance. In particular I am concerned that useing 1-h2 fro residual variance will make the algorithm anticonservate (see above) by underestimating the residual variance in the regression problem.

- Assess the calibration of PIPs for Sparsepro+ (e.g. Figure S1 in SuSiE manuscript)-- the AUC plots tell us that ranking variants by PIP is good, but it doesn't tell us that thresholding at some nominal PIP value controls the false positive rate. Good PIP callibration would go a long way in addressing my conerns about the choice of residual variance parameter.

**Minor points**

- Maybe a simpler enrichment analysis for the UKBB biomarkers would be (1) causal variants in this phenotype vs (2) causal variants discovered in other phenotypes. It would more clearly highlight that the enrichment of the tissue-specific annotation in the relvant biomarker is above and beyond the background level enrichment of enrichment across causal variants discovered in all phenotypes.

- Were causal variants defined as the top variant per credible set or all variants in the credible set?

- It is not clear to me which annotations are used in the UKBB biomarkers analysis. This should be clearly stated in 4.4 or methods.

- To clarify the comparison between polyfun and Sparsepro it may be good to (1) run Sparsepro with the prior inclusion probabilities derived from polyfun and (2) fit Sparsepro with the exact same annotations used in polyfun (without screening annotations based on significance first). (1) vs Sparsepro+ would demonstrate that Sparsepro+ is making better use of the annotation information. (2) vs Sparsepro+ would emphasize the benfit of selecting annotations based on Gtest.

**Have all data underlying the figures and results presented in the manuscript been provided?**

Reviewer #1: Yes

Reviewer #2: Yes

Reviewer #3: Yes

PLOS authors have the option to publish the peer review history of their article (what does this mean?). If published, this will include your full peer review and any attached files.

Reviewer #1: No

Reviewer #2: No

Reviewer #3: No

---

## [Decision Letter · Decision Letter 1]

29 Aug 2023

Dear Dr Zhang,

Thank you very much for submitting your Research Article entitled 'SparsePro: an efficient fine-mapping method integrating summary statistics and functional annotations' to PLOS Genetics.

The manuscript was fully evaluated at the editorial level and by independent peer reviewers. The reviewers appreciated the attention to an important topic but identified some concerns that we ask you address in a revised manuscript.

We therefore ask you to modify the manuscript according to the review recommendations. Your revisions should address the specific points made by each reviewer.

Yours sincerely,

Xiaofeng Zhu

Section Editor

PLOS Genetics

Xiaofeng Zhu

Section Editor

PLOS Genetics

Reviewer's Responses to Questions

**Comments to the Authors:**

Reviewer #1: Thanks for addressing the issues. I have the following follow-up questions:

1. SparsePro uses a different formulation, but the underlying model is same as SuSiE. The K effect groups are same as K single effects. So I suggest removing the discussion on 'Equivalence between the SuSiE IBSS algorithm and a paired mean field variational inference algorithm'. The main contributions of the manuscript lie in the functional annotation, hyperparameter estimation, and posterior summary.

2. FINEMAP provides credible sets in the output cred file.

3. In the supplementary note line 82, the authors said 'it might be challenging to find the appropriate threshold' for purity. However, it is important to note that the threshold for entropy (log(20)) is also arbitrary. The highly correlated variants could be more than 50 in complex regions. Does this threshold, 20, correspond to a purity level in your simulations? Could you summarize the purity for the output CSs? How does the result look like if SuSiE uses the corresponding purity filter?

4. I'm still unclear about the CSs around the boundary of central 1MB region. For a 3Mb window, it has 3 parts, left 1Mb, central 1Mb, right 1Mb. The result for the central 1Mb part is used. What about the CS with SNPs at the right end of the central 1Mb and the left end of the right 1Mb? How do you address the results?

5. Is the \\tau_\\beta same for all effect groups? SuSiE allows different effect priors for each single effect.

6. Extracting information from the large supplementary table S1 according to lines 100-107 is challenging. Consider presenting these results in a figure format or incorporating them into Figure 3 for better clarity.

7. What's the largest K used when fitting the SparsePro model in simulations and applications? What's the parameter setting for SuSiE, FINEMAP and PAINTOR?

Reviewer #2: I am positive about publishing this manuscript in PLOS Genetics.

I would like to apologize to the authors for having taken a long time before taking the time to read through their revision, I have tried to do it seriously when I had the time to do so. I think the authors answered my concerns, as well as the other reviewers' concerns, in a satisfactory way and put a substantial amount of work into improving the manuscript.

Reviewer #3: Uploaded as attachment.

**Have all data underlying the figures and results presented in the manuscript been provided?**

Reviewer #1: Yes

Reviewer #2: Yes

Reviewer #3: Yes

PLOS authors have the option to publish the peer review history of their article (what does this mean?). If published, this will include your full peer review and any attached files.

Reviewer #1: No

Reviewer #2: No

Reviewer #3: No

---

## [Decision Letter · Decision Letter 2]

11 Dec 2023

Dear Dr Zhang,

We are pleased to inform you that your manuscript entitled "SparsePro: an efficient fine-mapping method integrating summary statistics and functional annotations" has been editorially accepted for publication in PLOS Genetics. Congratulations!

Yours sincerely,

Gao Wang

Guest Editor

PLOS Genetics

Xiaofeng Zhu

Section Editor

PLOS Genetics

Comments from the reviewers (if applicable):

Reviewer's Responses to Questions

**Comments to the Authors:**

Reviewer #1: Thanks for the response. I don't have any additional concerns.

Reviewer #2: While I think that the points raised by the other reviewers are interesting from my perspective I still think that the manuscript is good enough for publication in PLOS Genetics.

Reviewer #3: I thank the authors for addressing the issues that were raised. The authors have answered my concerns and those of the other reviews. The paper makes an important contribution of providing a way to incorporate annotations into SuSiE-style fine-mapping. I support acceptance of the paper.

**Have all data underlying the figures and results presented in the manuscript been provided?**

Reviewer #1: Yes

Reviewer #2: Yes

Reviewer #3: Yes

PLOS authors have the option to publish the peer review history of their article (what does this mean?). If published, this will include your full peer review and any attached files.

Reviewer #1: No

Reviewer #2: No

Reviewer #3: No

**Data Deposition**

http://datadryad.org/submit?journalID=pgenetics&manu=PGENETICS-D-23-00072R2

**Press Queries**

---

## [Editor Report · Acceptance letter]

21 Dec 2023

PGENETICS-D-23-00072R2 

SparsePro: an efficient fine-mapping method integrating summary statistics and functional annotations 

Dear Dr Zhang, 

We are pleased to inform you that your manuscript entitled "SparsePro: an efficient fine-mapping method integrating summary statistics and functional annotations" has been formally accepted for publication in PLOS Genetics! Your manuscript is now with our production department and you will be notified of the publication date in due course.

With kind regards,

Zsofi Zombor

PLOS Genetics

On behalf of:
